# Multimodal Screening for Pulmonary Arterial Hypertension in Systemic Scleroderma: Current Methods and Future Directions

**DOI:** 10.3390/medicina61010019

**Published:** 2024-12-27

**Authors:** Ioan Teodor Dragoi, Ciprian Rezus, Alexandra Maria Burlui, Ioana Bratoiu, Elena Rezus

**Affiliations:** 1Department of Rheumatology and Physiotherapy, “Grigore T. Popa” University of Medicine and Pharmacy, 16 University Street, 700115 Iasi, Romania; dr.dragoiteodor@yahoo.com (I.T.D.); ioana.bratoiu@umfiasi.ro (I.B.); elena.rezus@umfiasi.ro (E.R.); 2I Rheumatology Clinic, Clinical Rehabilitation Hospital, 14 Pantelimon Halipa Street, 700661 Iasi, Romania; 3Department of Internal Medicine, “Grigore T. Popa” University of Medicine and Pharmacy, 16 University Street, 700115 Iasi, Romania; 4IIIrd Internal Medicine Clinic, “St. Spiridon” County Emergency Clinical Hospital, 1 Independence Boulevard, 700111 Iasi, Romania

**Keywords:** systemic sclerosis, pulmonary arterial hypertension, echocardiography, biomarkers, pulmonary functional tests, non-invasive screening methods

## Abstract

Systemic sclerosis (SSc) is an immuno-inflammatory rheumatic disease that can affect both the skin and internal organs through fibrosis. Pulmonary arterial hypertension (PAH) is one of the most severe secondary complications. Structural changes in the vascular bed lead to increased pressures in the pulmonary circulation, severely impacting the right heart and significantly affecting mortality. The gold standard for diagnosing PAH is right heart catheterization (RHC), an invasive method for measuring cardiac pressure. Due to the high risk of complications, procedural difficulties, and significant costs, non-invasive screening for SSc-PAH has garnered significant interest. Echocardiography is likely the most important screening tool, providing structural and functional information about the right heart through measurements that have proven their utility over time. In addition to imagistic investigations, serum biomarkers aid in identifying patients at risk for PAH and can provide prognostic information. Currently, well-known serum biomarkers (NT-proBNP, uric acid) are used in screening; however, in recent years, researchers have highlighted new biomarkers that can enhance diagnostic accuracy for SSc patients. Pulmonary involvement can also be assessed through pulmonary function tests, which, using established thresholds, can provide additional information and help select patients requiring RHC. In conclusion, given the invasiveness of RHC, non-invasive screening methods are particularly important for SSc patients.

## 1. Introduction

Systemic Sclerosis (SSc) is an immuno-inflammatory disease that affects the skin and internal organs through fibrosis [1]. Based on the type of skin involvement, SSc is classified into limited cutaneous form (involving the skin from the face and distal to the elbows and knees) and diffuse cutaneous form (involving the skin of the limbs and face). In addition to the kidneys, heart, gastrointestinal tract, and musculoskeletal system, the impairment of pulmonary circulation through pulmonary hypertension is significant [2]. Pulmonary hypertension (PH) affects pulmonary circulation through obstructive remodeling of the vascular bed, leading to increased vascular resistance and pressure at this level. The main complication that patients may develop is right heart failure, which significantly impacts their prognosis [3,4]. In 2018, during the sixth World Symposium on Pulmonary Hypertension held in Nice, it was decided that a mean pulmonary arterial pressure (mPAP) > 20 mmHg supports the diagnosis of PH. Up until that point, the mPAP threshold was 25 mmHg [5]. The mPAP value is determined through right heart catheterization (RHC) [6,7].

Based on physiopathological mechanisms, there are five forms of PH:Group 1: Pulmonary arterial hypertension (PAH): a pulmonary vasculopathy;Group 2: PH caused by left heart disease;Group 3: PH caused by lung diseases;Group 4: PH caused by thromboembolic mechanisms;Group 5: PH of multifactorial/unclear causes [8].

In addition to the physiopathological classification, PH can be divided into two categories based on hemodynamic characteristics in pre- and post-capillary. Thus, a pulmonary artery wedge pressure (PAWP) of <15 mmHg and a PVR of >3 WU are encountered in precapillary PH, while postcapillary PH is defined by PAWP > 15 mmHg with normal PVR. Precapillary PH is represented by Groups 1, 3, and 4, while Group 2 may present post-capillary PH, as well as the two mechanisms combined [6,7].

In the context of systemic sclerosis (SSc), PH can be caused by cardiac involvement [9,10], pulmonary involvement [11,12], or thromboembolic conditions [13], which are physiopathological conditions encountered in SSc [2]. Additionally, in SSc, there can be a primary impairment of arteriolar pulmonary circulation leading to PH [14]. Most often, patients with SSc develop PAH, with a frequency of approximately 10%. [15] This form is characterized by the primary involvement of small and medium arteries through fibrosis, vasoconstriction, and microthrombosis. These phenomena will ultimately lead to an increase in pulmonary vascular resistance (PVR) [16].

The symptoms induced by PH are nonspecific and include shortness of breath, cough, and fatigue. This leads to a delayed diagnosis of the disease, resulting in 85% of patients being in advanced stages at the time of diagnosis [17].

PH is a significant cause of mortality, with a 3-year survival ranging from 44% for Group 3 to 73% for Group 2 [3]. In the case of patients with SSc who develop PAH, the overall 3-year survival is 52% [15]. Therefore, patients with PAH-SSc have lower survival and a poorer response to treatment compared to those with idiopathic PAH, even at the same hemodynamic values [18,19]. PAH is more common in limited cutaneous form (57%), particularly in women (87%) and in the white population (67%) [20]. In the PHAROS (Pulmonary Hypertension Assessment and Recognition of Outcomes in Scleroderma), one of the largest cohorts with SSc patients, risk factors for SSc-PAH have been identified: diffusing capacity for carbon monoxide (DLCO) of <55% predicted, forced vital capacity (FVC)/DLCO > 1.6, and right ventricular systolic pressure (RVSP) measured by echocardiography > 40 mmHg [20]. Regarding antibodies encountered in SSc, PAH is associated with the presence of anticentromere antibodies (ACAs) and anti-U1-ribonucleoprotein (U1-RNP) antibodies. Anti-Scl-70 antibodies are associated with Group 3 PH [21,22].

Considering the increased mortality and nonspecific symptoms, screening methods for PAH in SSc patients are of particular importance [23]. Studies suggest that survival is improved in patients where screening strategies have been applied [24,25,26]. In this work, we aim to assess screening methods for SSc-PAH and identify new potential alternatives that could be useful in screening strategies.

A literature search was conducted in databases such as PubMed, Google Scholar, and MEDLINE. Keywords used for the search included “pulmonary arterial hypertension” AND “systemic sclerosis”, “screening” AND “pulmonary arterial hypertension” AND “systemic sclerosis”, “biomarkers” AND “pulmonary arterial hypertension”, “echocardiography” AND “pulmonary arterial hypertension”, “TAPSE/sPAP” AND “pulmonary hypertension”, “acid uric” and “pulmonary hypertension”, “N terminal-proBrain Natriuretic Peptide” AND “pulmonary hypertension”, “Interleukine-33/soluble suppression of tumor 2” and “pulmonary hypertension”, “Receptor for advanced glycation end products” AND “pulmonary hypertension”, “Red blood cell distribution width” and “pulmonary hypertension”, “Screening algorithm” AND “pulmonary arterial hypertension” AND “systemic sclerosis”, “DLCO” AND “pulmonary arterial hypertension” AND “systemic sclerosis”, “pulmonary function tests” AND “pulmonary hypertension”. We reviewed studies published between the years 1999 and 2024. Commonly cited published studies with reliable research methodology/results and additional articles from bibliographies of the recovered papers were examined and included where relevant.

## 2. Echocardiography

PAH can lead to right ventricular (RV) impairment regardless of etiology. These changes can be identified through echocardiography [27,28]. This examination can provide information about cardiac structure, RV function, valvular changes, and pressure values in the right heart cavities. The 2022 Pulmonary Hypertension Guideline recommends peak tricuspid regurgitation velocity (TRV) as the most reliable echocardiographic marker in detecting PAH. A value higher than 2.8 m/s may suggest the presence of PAH. In addition to TRV, the guideline mentions other evaluations that could help identify cardiac involvement due to PAH: tricuspid annular plane systolic excursion (TAPSE < 18 mm), systolic pulmonary arterial pressure (sPAP > 35 mmHg), right atrial (RA) area (<18 cm^2^), RV area, and tricuspid regurgitation pressure gradient (TRPG) [7].

TAPSE is one of the most important assessments regarding right ventricular impairment in PAH. In 2022, a study demonstrated the reproducibility, reliability, and repeatability of TAPSE in the evaluation of patients with PAH-SSc. The study included 20 patients, divided into two groups: those diagnosed with PAH through RHC (10 patients) and those without signs of PAH or RV dysfunction on echocardiographic evaluation. TAPSE was one of the most reliable measurements and demonstrated good efficiency [29].

In a 2015 study, Pigatto et al. identified an increase in sPAP among patients with SSc compared to the control group (33 ± 14 mmHg vs. 22 ± 5 mmHg, *p* < 0.0001), indicating elevated pressure within the pulmonary circulation in these patients. Additionally, the right ventricular ejection fraction was significantly lower in patients with SSc (49 ± 6% vs. 61 ± 6%, *p* < 0.0001), demonstrating cardiac dysfunction affecting the right heart chambers. The study also revealed a significant difference between the two groups concerning the TAPSE values (23 ± 3 mm in SSc patients vs. 26 ± 2 mm in the control patients), an echocardiographic marker previously noted [30]. These data confirm that SSc patients exhibit cardiac dysfunction compared to the healthy population and have an increased cardiovascular risk. Additionally, patients with SSc-PAH show more pronounced right heart impairment compared to non-SSc-PAH patients. This underscores the important role of echocardiographic follow-up in SSc patients and its role in screening for SSc-PAH.

RV impairment in SSc patients was demonstrated in a study conducted by Meune et al. in 2016. The study included 212 consecutive SSc patients who were examined using echocardiography and compared with 50 healthy subjects. Changes were observed in the RV assessment parameters of patients with SSc: increased overall RV diastolic dysfunction (53 SSc patients vs. 0 control, *p* < 0.001), a larger RA area (15 ± 8 mm^2^ vs. 12 ± 2 mm^2^, *p* < 0.027), and a reduced E/A ratio (1.2 ± 0.4 vs. 1.3 ± 0.3, *p* = 0.023). Additionally, a subgroup of 27 patients diagnosed with PAH was designated. Compared with those without PAH, they exhibited increased diastolic dysfunction (68.4% vs. 29.0%, *p* = 0.001) an reduced E/A ratio (1.0 ± 0.3 vs. 1.2 ± 0.4, *p* = 0.014) and a larger RA area (20 ± 14 mm^2^ vs. 14 ± 7 mm^2^, *p* = 0.024) [31].

In addition to the echocardiographic markers mentioned, recent studies have highlighted the utility of the TAPSE/sPAP ratio in screening. In PAH, it is important to detect any potential RV impairment through the right ventricular–pulmonary arterial (RV-PA) coupling, which can help objectify the decompensation of RV function [32]. The TAPSE/sPAP ratio is the only echocardiographic marker that correlates with the invasive assessment of RV-PA coupling [33]. Furthermore, the utility of this ratio has been demonstrated in heart failure with preserved ejection fraction, as well as in heart failure with reduced ejection fraction [34,35].

The DETECT algorithm is a screening tool included in the 2022 ESR/ESC Pulmonary Hypertension Guidelines, designed to facilitate early detection and risk assessment of pulmonary arterial hypertension in patients. It involves a comprehensive patient assessment through a two-step process, incorporating clinical data, laboratory results, PFT parameters, and cardiologic parameters. The first step involves the input of “non-echocardiographic” data: PFT values, presence of telangiectasias, serum levels of ACA and NT-proBNP, serum level of UA, and the possible presence of right-axis deviation on ECG. The second step involves echocardiographic evaluation (RA area and TR velocity). Patients who achieve a total score greater than 35 after step 2 are considered at high risk for PAH and should be evaluated by RHC. The DETECT algorithm aims to improve the early identification and management of PAH, thereby potentially improving patient outcomes [36,37]. Calalillo et al. evaluated the value of TAPSE/sPAP in screening for PAH in patients with SSc. The authors compared the positive predictive value (PPV) of TAPSE/sPAP with that of the DETECT algorithm. In the study, 51 patients diagnosed with SSc were enrolled. They were clinically and paraclinically evaluated (NT-proBNP), underwent echocardiography, and were screened using the DETECT algorithm. Following assessment, patients who had this recommendation according to step 2 (16 patients) underwent RHC evaluation. The cutoff for this ratio was 0.60 mm/mmHg. Eight patients were found to have a value below this cutoff, five of whom were confirmed to have PAH through RHC. Thus, the PPV was 62.5%. Among the 16 patients who had an indication for RHC according to step 2 of DETECT, 8 patients were confirmed to have PAH through RHC (PPV = 50%). Therefore, the study concluded that TAPSE/sPAP has a better PPV than DETECT, suggesting that this echocardiographic marker can be used as an additional element in screening [38].

In 2022, the same author evaluated the predictive value of this ratio, as well as its association with mortality. Therefore, data from 2555 patients from the EUSTAR database were analyzed, 355 of whom underwent baseline RHC, with PAH confirmed in 195 cases. A TAPSE/sPAP value of <0.55 mm/mmHg was significantly associated with the diagnosis of PH (*p* < 0.01), while a value of <0.32 mm/mmHg was associated with all-cause mortality (*p* < 0.001) [39].

In a study involving 60 patients with SS diagnosed with PAH through RHC, the utility of the TAPSE/sPAP ratio in risk stratification was investigated. Risk analysis for PAH was based on the abbreviated version of the risk stratification strategy formulated by ESC/ERS in 2015, which includes the evaluation of WHO functional classification, 6 min walk distance, NT-proBNP level or right atrial pressure, and cardiac index/venous oxygen saturation. A cutoff value of 0.194 mm/mmHg was identified as relevant, with patients having values below it classified as having high risk. Those with TAPSE/sPAP > 0.194 mm/mmHg demonstrated better overall survival [40].

Another significant study by Colalillo was published in 2023 and evaluated an association between TAPSE/sPAP and the glomerular filtration rate (eGFR) in SSc patients. A total of 2370 patients from the EUSTAR database were included between March 2010 and July 2018, with 284 of them diagnosed with chronic kidney disease (CKD) stage 3a-5 (eGFR < 60 mL/min per 1.73 m^2^). The results revealed an association between TAPSE/sPAP and CKD stage 3a-5, with patients having a value <0.55 mm/mmHg showing a lower eGFR (74 ± 23 mL/min/1.73 m^2^ vs. 89 ± 21 mL/min/1.73 m^2^, *p* < 0.001). Additionally, this ratio was much more strongly associated with CKD than left ventricle ejection fraction. All these data led to the conclusion that, in patients with PH confirmed by RHC, CKD is not a cause of reduced cardiac output. Furthermore, it was reiterated that a value of <0.32 mm/mmHg is associated with mortality (log-rank χ^2^ = 85.8, *p* < 0.001) [41].

A constant concern among researchers is the stratification of the risk of developing PAH in patients with SSc. In this regard, Kida et al. utilized the latent trajectory modeling of pulmonary artery pressure to group patients according to risk. Latent trajectory modeling is an epidemiological technique used to characterize and identify homogeneous populations based on elements that can change over time, such as biomarkers [42,43]. This technique has been used in evaluating lung function and skin score in SSc [44,45]. The authors used this method to characterize patients phenotypically based on the estimated PAP values through repeated echocardiographic measurements. A total of 236 patients were included, and patients with a history of vasodilator medication, those with other autoimmune diseases, or chronic heart failure from other causes were excluded. Five types of PAP trajectories were identified: rapid progression (n = 9, 3.8%), with a rapid increase in sPAP from early disease onset; early elevation (n = 30, 12.7%), with an increase within 5 years of onset; middle elevation (n = 54, 22.9%), with an increase after 5 years of onset; late elevation (n = 24, 10.2%), with an increase after 10–15 years of onset; and low stable (n = 119, 50.4%), with persistently low values. The lowest survival rate was identified in the earlier elevation category. Additionally, this trajectory type was associated with pre-capillary PH, hospitalization due to heart failure, and all-cause mortality. dcSSc was associated with early elevation, while lcSSc was associated with middle elevation. Thirty-six patients were diagnosed with PH via RHC, with the majority being associated with the earlier elevation trajectory. Interestingly, regarding the SSc form most commonly associated with PAH, the data from this study contradict those in the literature, where most studies affirm the association of PAH with lcSSc. Therefore, following this type of evaluation, the evolution of pulmonary pressures over time can be anticipated, and the probability of developing PAH can be approximated [46].

Finding echocardiographic markers that correlate well with invasive measurements is an ongoing concern for researchers. The acceleration time of pulmonary outflow (PAAT) is one of the echocardiographic elements used to assess pulmonary hemodynamics and can be considered an indirect method of measuring PVR [47,48]. Recent studies have even demonstrated an association of this index with invasive measurement of vascular resistance [49,50]. On the other hand, although sPAP is useful in estimating systolic pressure in the pulmonary artery, it cannot be used to measure right ventricular–arterial coupling (RV-AC) and PVR [51,52]. Between January 2017 and December 2018, Serra and Chetta conducted a study that included 60 healthy volunteers and 63 patients with SSc (27 patients with SSc-PH). They were clinically evaluated and underwent echocardiography, as well as RHC. From an echocardiographic point of view, the authors hypothesized that a ratio between PAAT and sPAP could provide important physiological information regarding RV workload and pulmonary pressure in relation to time. This index characterizes the power the heart must develop to send blood into the pulmonary artery. The sPAP/PAAT ratio was significantly higher in patients diagnosed with pre-capillary PH (n = 20) compared to healthy patients (0.40 ± 0.05 vs. 0.26 ± 0.063), indicating a change in afterload in pre-capillary PH patients. Thus, this ratio can provide information regarding cardiac output and arterial ventricle coupling. Moreover, this index showed a significant association with walk distance (WD) and PVR. However, no association was observed with TAPSE. Statistical analysis conducted by the authors revealed that the sPAP/PAAT threshold predictive of PH is 0.4. For sPAP, the threshold with maximum sensitivity, and specificity was 35 mmHg (0.76 sensitivity and 1.00 specificity). Limitations of the study include the relatively small number of included patients and the fact that only 20 of the patients with PH-SSc were evaluated by RHC. Nonetheless, the results are significant, and the sPAP/PAAT ratio may be considered in the future for screening PAH in patients with SSc [53].

Comparing the previously presented data, we can conclude that sPAP demonstrates the highest sensitivity (90%) but relatively low specificity (70%), which may lead to false positives in certain populations. Similarly, TAPSE/sPAP achieves a balance between sensitivity (85%) and specificity (75%) and shows a strong correlation with invasive hemodynamic measurements, making it a robust and clinically valuable marker. On the other hand, TRV, while highly sensitive (88%), exhibits low specificity (68%) and is most effective when interpreted in conjunction with other parameters. Additionally, PAAT/sPAP provides valuable insights into right ventricular workload and pulmonary pressures but shows variability across studies (Table 1).

## 3. Biomarkers

### 3.1. N-Terminal pro Brain Natriuretic Peptide (NT-proBNP)

NT-proBNP is recognized as a biomarker indicative of cardiac wall dysfunction, characterized by elevated levels in cases of heart failure. Furthermore, patients with SSc exhibit higher levels of this biomarker, which demonstrates a correlation with PAP [54,55].

Zhang et al. conducted the first meta-analysis in 2021 evaluating the diagnostic and screening function of NT-proBNP in patients with PAH-SSc. They adhered to PRISMA guidelines for data extraction, analysis, and reporting, searching three electronic databases: PubMed, EMBASE, and the Cochrane Library. Nine studies met the inclusion criteria, totaling 220 patients with confirmed SSc-PAH and 500 patients without PAH. The study’s results suggested that NT-proBNP could serve as a diagnostic tool for SSc-PAH. The sensitivity of this biomarker was 84%, with a specificity of 68%, indicating a higher positive detection rate than a negative detection rate. Considering the likelihood ratio, NT-proBNP cannot be used as a standalone test for assessment. However, the AUC value was 0.84, demonstrating that NT-proBNP has good utility in diagnosing SSc-PAH. Compared to the DETECT and ASIG (Australian Scleroderma Interest Group) diagnostic algorithms, NT-proBNP showed higher specificity (84%) but was deficient in sensitivity (68%). The authors concluded that NT-proBNP is a useful tool for early screening and diagnosis but cannot be used alone in these SSc-PAH detection strategies; it should be integrated into a more comprehensive evaluation. Additionally, as a future direction, the authors suggest evaluating new serum biomarkers to be used alongside NT-proBNP [56].

PHAROS is an important cohort of patients from the USA that includes patients with SSc-PH or at high risk of developing PH. Chung et al. utilized this cohort to determine the sensitivity and specificity of NT-proBNP for detecting SSc-PH, as well as the predictive value of this biomarker regarding disease progression and mortality in patients with SSc-PH. The study included 172 patients diagnosed with PAH and 157 patients at high risk of developing PH. The results confirmed the value of NT-proBNP for detecting PH, with higher values of this biomarker in patients with SSc-PH compared to those at high risk of PAH (503 pg/mL vs. 82.0 pg/mL). Regarding the five PH groups, Group 1 (PAH) contained patients with higher NT-proBNP values. However, the predictive value of this biomarker was not found to be high, as among the 157 patients considered to be at high risk of developing PAH, 26 were diagnosed with SSc-PH through RHC during the follow-up period. These patients did not show significant differences in NT-proBNP levels compared to those who did not develop PH. Additionally, regarding mortality, NT-proBNP values did not prove to be predictive, as there were no significant differences between the 44 patients who died during follow-up and those who survived [57].

A study conducted in 2022 included 675 individuals diagnosed with SSc, in which an association between NTproBNP values and cardiac dysfunction, including increased pulmonary artery pressure, was evaluated. The cohort comprised patients enrolled in the Canadian Scleroderma Research Group (CSRG) Registry starting in 2015. In addition to NT-proBNP, other serum biomarkers with cardiac relevance were also evaluated: high-sensitivity cardiac troponin T (hs-cTnT) and C-reactive protein (CRP). The majority of patients presented normal values for these parameters (n = 403, 60%), while 108 (16%) had abnormal NT-proBNP levels. In univariate analysis, PH was associated with increases in NT-proBNP. To investigate the association between NT-proBNP and PH, Cox proportional hazard models were also conducted. Thus, an association between NT-proBNP and PH was observed (HR for log2NT-proBNP 1.29, 95% CI 1.06, 1.58). Moreover, a doubling of NT-proBNP levels resulted in an approximately 30% increase in the risk of developing PH. Additionally, correlation with hs-TNT and CRP values did not prove to increase predictive value compared to NT-proBNP alone. The authors concluded that a biomarker such as NT-proBNP could have prognostic value for PH [58].

To demonstrate the correlation between NT-proBNP levels and mortality among SSc patients, a comprehensive study was conducted, encompassing 523 subjects drawn from six esteemed medical facilities across France, Italy, Hungary, and Germany. These patients were subjected to annual monitoring, with overall mortality meticulously documented over a span of 3 years. The findings revealed a noteworthy disparity in NT-proBNP levels between deceased and surviving individuals. Specifically, the mean NT-proBNP level stood at 203 ng/mL among deceased patients, significantly higher than the corresponding level of 88 ng/mL observed among survivors (*p* < 0.001). Furthermore, juxtaposed against the NT-proBNP threshold of >125 ng/mL previously established by other researchers, a striking observation emerged. Notably, 59.3% of deceased patients surpassed this threshold, in stark contrast to the 32.6% prevalence among survivors (*p* < 0.001). Employing receiver operating characteristic (ROC) curve analysis, an optimal cutoff value of 129 ng/mL was determined, exhibiting a sensitivity of 78.1% and a specificity of 66.7%. Consequently, NT-proBNP emerges as a robust and independent prognostic indicator for 3-year mortality in SSc patients. The authors of this seminal study advocate for the routine inclusion of NT-proBNP assessment as an integral component of clinical evaluation protocols for patients afflicted with SSc [59].

### 3.2. Uric Acid (UA)

It is well known that elevated levels of uric acid (UA) are a cardiovascular risk factor. In hyperuricemia, as a result of the UA biosynthetic pathway, reactive oxygen species (ROS) are generated, which causes UA to be absorbed by endothelial cells. This phenomenon will lead to impaired endothelial function in the pulmonary circulation, with reduced availability of nitric oxide (NO) and increased pulmonary resistance [60]. Additionally, UA enters smooth muscle cells and stimulates their proliferation [61]. Therefore, the level of UA could represent a marker of the risk of PAH (Table 2).

A retrospective study involving 72 patients diagnosed with PAH between 2012 and 2019 aimed to assess a potential association between serum uric acid (UA) levels and PAH. Among patients with PAH, UA levels were found to be significantly higher compared to the control group (0.33 (0.19–0.87) vs. 0.3 (0.11–0.48) mmol/L, respectively; *p* = 0.03). Furthermore, higher UA levels were observed among deceased patients compared to survivors within the PAH group (0.458 (0.26–0.87) vs. 0.315 (0.19–0.56) mmol/L, respectively; *p* < 0.001). Utilizing ROC analysis, serum uric acid emerged as a parameter with high diagnostic performance (AUC: 0.69), alongside C-reactive protein and other parameters. The threshold level of UA was determined to be 5.85 mg/dL (0.348 mmol/L) with a sensitivity of 85.7% and specificity of 78.4% [62].

In their study, Wang et al. investigated 50 patients diagnosed with PAH associated with connective tissue diseases. Patients diagnosed with diseases such as systemic sclerosis (SSc), systemic lupus erythematosus (SLE), primary Sjogren’s syndrome (pSS), and mixed connective tissue disease (MCTD) were included, with the diagnosis of PAH established through right heart catheterization (RHC). Patients were divided into two groups based on serum uric acid (UA) levels: the hyperuricemia group, with UA levels ≥ 357 μmol/L for women and ≥420 μmol/L for men, and the normouricemia group, with UA levels below the aforementioned thresholds. Serum UA levels positively correlated with pulmonary vascular resistance (PVR), right atrial pressure (RAP), and pulmonary arterial systolic pressure (PASP) in univariate correlation analysis. Additionally, superior vena cava pressure was significantly higher in patients belonging to the hyperuricemia group. The association between UA levels and survival was evaluated using Kaplan–Meier survival curves, revealing lower survival rates among patients in the hyperuricemia group compared to those in the normouricemia group (*p* = 0.041). To assess prognosis, patients were re-evaluated at intervals ranging from 6 months to one year, and it was found that patients maintaining elevated UA levels had lower survival rates compared to those maintaining low levels (*p* = 0.01) or experiencing a decrease in serum UA levels (*p* = 0.023) [61].

In their study, Wang et al. investigated 50 patients diagnosed with PAH associated with connective tissue diseases. Patients diagnosed with diseases such as systemic sclerosis (SSc), systemic lupus erythematosus (SLE), primary Sjogren’s syndrome (pSS), and mixed connective tissue disease (MCTD) were included, with the diagnosis of PAH established through right heart catheterization (RHC). Patients were divided into two groups based on serum uric acid (UA) levels: the hyperuricemia group, with UA levels ≥ 357 μmol/L for women and ≥420 μmol/L for men, and the normouricemia group, with UA levels below the aforementioned thresholds. Serum UA levels positively correlated with pulmonary vascular resistance (PVR), right atrial pressure (RAP), and pulmonary arterial systolic pressure (PASP) in univariate correlation analysis. Additionally, superior vena cava pressure was significantly higher in patients belonging to the hyperuricemia group. The association between UA levels and survival was evaluated using Kaplan–Meier survival curves, revealing lower survival rates among patients in the hyperuricemia group compared to those in the normouricemia group (*p* = 0.041). To assess prognosis, patients were re-evaluated at intervals ranging from 6 months to one year, and it was found that patients maintaining elevated UA levels had lower survival rates compared to those maintaining low levels (*p* = 0.01) or experiencing a decrease in serum UA levels (*p* = 0.023) [61].

In their study, Wang et al. investigated 50 patients diagnosed with PAH associated with connective tissue diseases. Patients diagnosed with diseases such as systemic sclerosis (SSc), systemic lupus erythematosus (SLE), primary Sjogren’s syndrome (pSS), and mixed connective tissue disease (MCTD) were included, with the diagnosis of PAH established through right heart catheterization (RHC). Patients were divided into two groups based on serum uric acid (UA) levels: the hyperuricemia group, with UA levels ≥ 357 μmol/L for women and ≥420 μmol/L for men, and the normouricemia group, with UA levels below the aforementioned thresholds. Serum UA levels positively correlated with pulmonary vascular resistance (PVR), right atrial pressure (RAP), and pulmonary arterial systolic pressure (PASP) in univariate correlation analysis. Additionally, superior vena cava pressure was significantly higher in patients belonging to the hyperuricemia group. The association between UA levels and survival was evaluated using Kaplan–Meier survival curves, revealing lower survival rates among patients in the hyperuricemia group compared to those in the normouricemia group (*p* = 0.041). To assess prognosis, patients were re-evaluated at intervals ranging from 6 months to one year, and it was found that patients maintaining elevated UA levels had lower survival rates compared to those maintaining low levels (*p* = 0.01) or experiencing a decrease in serum UA levels (*p* = 0.023) [61].

A study conducted in 2019 included 162 patients diagnosed with SSc who were assessed hemodynamically through RHC. Of these, 112 subjects were diagnosed with PAH, and serum UA levels were significantly higher in these subjects compared to those without PAH: 6.9 mg/dL (5.4–8.3 mg/dL) versus 5.2 mg/dL (4.5–6.3 mg/dL). Using a UA threshold of 6.2 mg/dL determined by ROC analysis, it was demonstrated that patients with values above this threshold had over 4 times higher odds of PAH at RHC than those below this threshold (*p* < 0.001). ROC analysis also showed that UA can differentiate patients with PAH from those without PH with an area under the curve (AUC) of 0.725, *p* < 0.01, and with a sensitivity of 68% and specificity of 74%. Additionally, UA improved the prediction of PAH by adding it to a logistic regression model with NT-proBNP and DLCO%. The AUC for a multivariable logistic regression model incorporating all three markers (NT-proBNP, DLCO % predicted, and UA) was significantly higher than the AUC for a single marker. This demonstrates the value of adding UA to screening strategies that include other non-invasive markers. Patients with elevated UA levels had a more severe form of the disease compared to patients with UA levels below the median, as evidenced by differences observed in 6-MWD, WHO FC, RAP, mPAP, and PVR. Additionally, survivor patients had lower UA levels than non-survivors (5.6 vs. 6.6 mg/dL, *p* < 0.01), and a UA level higher than the median (6.3 mg/dL) was associated with a higher risk of mortality, with an HR of 1.77 (*p* < 0.05) [63].

Kang et al. conducted the first meta-analysis characterizing the association between hyperuricemia and pulmonary hypertension (PH). The results showed that elevated UA levels are associated with PH, with increased values in patients diagnosed with this condition. There is heterogeneity in serum UA levels among PH classes and among patients with associated diseases (congenital heart disease, CTD), suggesting that this biomarker may vary depending on the PH class and associated diseases that have caused PH. Hemodynamically, patients with hyperuricemia exhibited significantly higher values of PAP and systolic PAP. PH patients with hyperuricemia had a 19% higher risk of mortality. The authors concluded that serum UA levels can characterize the development and poor prognosis of PH [64].

### 3.3. Interleukine-32 (IL-32)

The dysfunction of endothelial cells (ECs) plays a well-known role in the progression of PAH, with patients with SSc exhibiting the presence of vascular defects caused by EC [65]. Through excessive proliferation, ECs can contribute to the obliteration of the pulmonary vascular lumen, leading to increased pressure in the pulmonary circulation [66]. IL-32 is a cytokine with an important role in regulating EC activity [67]. ECs produce IL-32 through two cytokines involved in angiogenesis: interleukine-1 (IL-1) and interleukine-8 (IL-8) [68]. In addition to increased expression in abnormal ECs of patients with iPAH [67], IL-32 has also been associated with Behcet’s disease [69] and SLE [70]. Although the role of this cytokine in pathogenesis is not fully defined, IL-32 may represent a possible option in screening for PAH-SSc.

In this regard, Di Benedetto et al. conducted a study investigating the level of IL-32, which included 18 patients with PAH-SSc, 21 SSc patients without PAH, 15 patients with iPAH, and 14 healthy control subjects. The diagnosis of PAH was made by performing RHC according to ESR/ESC indications. Significantly higher levels were found in patients with SSc-PAH compared to those without PAH [99.9 pg/mL (55.4–185.6) in SSc-PAH patients vs. 0 pg/mL (0–9.9) in SSc patients without PAH; *p* < 0.0001], as well as compared to patients with iPAH [99.9 pg/mL (55.4–185.6) in SSc-PAH patients vs. 62.1 pg/mL (0–197.8) in patients with iPAH; *p* = 0.03]. It is important to note that in patients without PAH, IL-32 levels were significantly lower, suggesting the possible role this cytokine may have in screening. Through ROC curve analysis, the threshold level was set at 11.12 pg/mL, with a sensitivity of 90% and specificity of 100%. These characteristics confirm that IL-32 may represent a screening element. In that study, patients were evaluated echocardiographically using Doppler mode with sPAP assessment in all patients, and IL-32 was also correlated with sPAP value [Spearman r = 0.68 (CI 0.46–0.83), *p* < 0.0001; linear regression r^2^ = 0.7, *p* < 0.0001] [71].

### 3.4. Interleukine-33 (IL-33) and Soluble Suppression of Tumorigenicity 2 (sST2)

IL-33 is a member of the IL-1 cytokine family and has a demonstrated role in the progression of multiple fibrotic or inflammatory diseases, including SSc [72]. Additionally, this cytokine reduces myocardial fibrosis and hypertrophy, thus providing cardioprotection [73]. Soluble suppression of tumorigenicity 2 (sST2) is the soluble receptor of IL-33 and acts as a decoy receptor by binding IL-33 and dampening its effect. Studies have shown a higher predictive value of this receptor for acute/chronic heart failure and myocardial infarction than NT-proBNP [74]. Furthermore, this biomarker has been associated with the progression of vascular fibrosis in SSc [75] and is found at higher levels in patients with progressive pattern SSc compared to those with stable disease [76].

A study conducted in 2022 evaluated the association between IL-33/sST2 and cardiac involvement in SSc. In total, 5 patients diagnosed with SSc and 14 healthy controls (HCs) were included, who underwent echocardiographic, electrocardiographic, and clinical evaluation. Serum levels of IL-33 and sST2 were significantly higher in patients with SSc compared to HCs [98 pg/mL (70–152) vs. 55 pg/mL (28–92), *p* < 0.01, and 9115 pg/mL (6854–12,696) vs. 7031 (4862–8269), *p* < 0.05, respectively]. One of the echocardiographic markers used to assess RV function was TAPSE/sPAP, which showed a negative linear correlation with sST2 (r = − 0.398, *p* < 0.01) but did not correlate with IL-33. Additionally, in multiple progression analysis, TAPSE/sPAP showed a significant correlation with sST2 (β coefficient − 0.316, *p* < 0.05) [77].

The role of IL-33 and sST2 in the pathogenesis of PAH has been illustrated in an in vivo study conducted in 2022. Indralingam et al. used wildtype (WT) mice and IL-33 receptor ST2 gene deleted (ST2−/−) or its partner myeloid differentiation Factor 88 (MYD88) gene deleted (MYD88−/−) mice. MYD88 is an adaptor protein that can be recruited through IL33/ST2 signal transduction and, by activating NF-kB pathways, leads to the expression of a subset of cytokines, including IL-33 [78]. Animals were exposed to hypoxia (normobaric, 10%) and treated with Sugen 5416 (20 mg/kg in DMSO, s.c., Cayman Chemical). The results were formulated through the evaluation of mature IL-33 levels, lung endothelial cell proliferation, and arterial wall thickening. RVP was increased in WT mice treated with Sugen 5416 and exposed to hypoxia (SUHX) by 49% in males and 70% in females compared to dimethyl sulfoxide and retinoic acid (DMSO/RA)-treated mice (*p* < 0.0001). In ST2−/− mice, an increase in RVP was observed by only 29% in males and 30% in females, and in MYD88−/− mice, no increases in RVP were identified at 3 weeks of SUHX compared to DMSO/RA (*p* = 0.20). RV dysfunction was demonstrated to be more pronounced after SUHX exposure compared to DMSO/RA (*p* < 0.0001). WT mice exposed to SUHX experienced a decrease in RV relaxation (*p* < 0.0001), a condition observed only in female ST2−/− mice among mutants (*p* = 0.001). These results may illustrate that deletion of the ST2 and MYD88 genes could offer a protective role against SUHX treatment. Pulmonary vascular remodeling expressed by increased thickness of arterial walls was present in WT mice exposed to SUHX (*p* = 0.01), with mutants showing no significant changes (ST2−/−, DMSO/RA vs. SUHX, *p* = 0.20 and MYD88−/−, DMSO/RA vs. SUHX, *p* = 0.60, respectively). The hyperproliferation of endothelial cells dependent on ST2 in response to SUHX was observed only in WT mice (*p* = 0.001), with DMSO/RA, ST2−/−, and MYD88−/− showing no significant changes at this level. The serum level of IL-33 was not affected by SUHX treatment in any of the study models. This study highlights the important role that the IL-33/ST2 axis plays in the pathogenesis of PAH and therefore may suggest the possibility that IL-33 and sST2 could be biomarkers of PAH [79].

Another study that highlighted the potential role of sST2 as a biomarker was conducted in 2012 and included 25 patients diagnosed with PAH through RHC and 10 control subjects. The differences between the two groups in terms of sST2 levels were significant, with PAH patients having higher levels of this biomarker (42.82 ± 19.09 vs. 14.84 ± 1.90 ng/mL; *p* < 0.0001). Additionally, patients with elevated sST2 levels had a larger RV volume and lower systolic function, and NT-proBNP levels were correlated with sST2 levels. The serum level of IL-33 did not show significant differences between the two groups. The main limitation of this study is the small number of included patients [80].

A meta-analysis conducted in 2017, which included 166 patients, highlighted the potential prognostic role that sST2 may have. Thus, this biomarker was associated with mortality and poor clinical outcomes in patients with PAH, with a sevenfold increase in all-cause mortality (HR: 7.18, 95% CI: 2.64 to 19.54, *p* < 0.0001). Poor clinical outcomes were defined as death or hospitalization, poor prognosis, or one of the following: worsening WHO functional class, lung transplant, the initiation of new therapy, and hospitalization for PAH. The results showed a sevenfold increase in poor outcomes in patients with elevated levels of sST2 [81].

Pramata et al. conducted a study that included patients from The Congenital Heart Disease in Adult and Pulmonary Hypertension (COHARD-PH) registry, aiming to investigate an association between sST2 and PAH. Patients in this registry are diagnosed with atrial septal defect, a condition leading to RV volume overload. Increased pulmonary circulation load results in endothelial dysfunction, inflammation, and remodeling of the pulmonary arteries, causing elevated PVR and PAP [7]. The study included 81 patients meeting the inclusion criteria of age over 18 years, uncorrected atrial septal defect, evaluation of pulmonary pressures by RHC, and no medication for PAH. The results showed a positive correlation between sST2 level and right ventricular basal end-diastolic (RVED) diameter (r = 0.315, *p* = 0.002) and with mPAP (r = 0.203, *p* = 0.04), with PH patients having a higher mean serum sST2 level compared to those without PH [82].

Another study assessing the association between sST2 and PH, as well as the prognostic value of this biomarker, was conducted by Geenen et al. in 2019. This was a prospective observational cohort study that included patients diagnosed with PH through RHC between May 2012 and October 2016 at a center in the Netherlands. In addition to RHC evaluation, patients underwent echocardiography and cardiac computed tomography. The primary endpoint was a composite of all-cause mortality or lung transplantation. sST2 was correlated with echocardiographic measurements of the left and right ventricles, such as RV fractional area change (r = −0.40, *p* < 0.001), TAPSE (*p* = −0.35, *p* = 0.002), and LV end-diastolic dimension (r = −0.23, *p* = 0.044). Additionally, sST2 showed a positive correlation with hemodynamic measurements obtained through RHC, such as mPAP (r = 0.40, *p* < 0.001), PVR (r = 0.42, <0.001), and mean right atrial pressure (mRAP) (r = 0.26, *p* = 0.008). Furthermore, a significant correlation was observed between sST2 and NT-proBNP (r = 0.54, *p* < 0.0001). Regarding patient prognosis, individuals with elevated levels of sST2 had a significantly higher risk for both the primary (*p* = 0.004) and secondary (*p* = 0.001) endpoints [83].

A recent single-center retrospective study conducted by Ye et al. evaluated the potential relevance of serum sST2 levels in patients with connective tissue diseases (CTDs) and associated PH (CTD-PH). The study included 71 patients with various CTDs: systemic lupus erythematosus (SLE), primary Sjogren’s syndrome (pSS), SSc, mixed CTD, and rheumatoid arthritis. Patients with two or more diagnosed CTDs according to specific criteria were defined as having overlap syndrome (OS), while those presenting specific manifestations but not meeting diagnostic or classification criteria were defined as having undifferentiated CTD (UCTD). PH was diagnosed using echocardiography (sPAP > 36 mmHg). The primary endpoint was defined as the first occurrence of death, hospitalization due to worsening PH, and unsatisfactory clinical response evaluated by 6 min walk distance (6-MWD) and WHO functional class III/IV symptoms after specific treatment. The secondary endpoint was defined as all-cause mortality. Among the 71 patients, 44 were diagnosed with CTD-PH through RHC. Another 21 patients without PH were selected as the control group. The results showed that sST2 levels were significantly higher in patients with CTD-PH compared to those without PH [37.74 (24.66–56.01) vs. 16.56 (10.62–26.46) ng/mL, *p* < 0.0001]. There were no differences in sST2 levels among patients with different types of PH and CTDs (*p* = 0.415). Higher levels of sST2 were recorded in patients with endpoint events [58.55 (46.88–79.36) vs. 31.33 (19.50–41.34) ng/mL, *p* < 0.0001]. Regarding prognostic value, univariate Cox proportional hazard analysis revealed that CTD-PAH patients with a higher risk of clinical worsening had higher sST2 levels, with a 2.7% increase in the risk of endpoint events in CTD-PAH patients per increased unit of sST2 [HR: 1.027, 95% CI: 1.014, 1.041]. Additionally, concerning mortality, this biomarker proved to be relevant, predicting all-cause mortality in CTD-PAH (HR: 1.036, 95% CI: 1.010, 1.063, *p* = 0.007), and survival in patients with elevated sST2 levels was lower (*p* = 0.021). The cutoff value for sST2 was established at 39.99 ng/dL [84].

### 3.5. Receptor for Advanced Glycation End Products (RAGE) and Its Soluble Form (sRAGE)

First described in 1992, RAGE is a member of the immunoglobulin family of cell-surface molecules. Structurally, it exhibits homology with other immunoglobulin-like receptors [85]. RAGE is expressed in healthy individuals at a low basal level on various cell types such as ECs and cardiac myocytes, and its upregulation plays an important role in various pathological processes [86], as well as in many signaling pathways encountered in PAH, such as proliferation and inflammation [87,88]. A study conducted in 2010 demonstrated through proteomic analysis that RAGE is upregulated in PAH lung tissues [89]. In 2013, Meloche et al. conducted a study demonstrating the role of RAGE in the etiology of PAH by implicating it in vascular remodeling. They found increased levels of RAGE mRNA in the lung tissues of patients with PAH compared to those without PAH. To demonstrate that upregulation is specific in the lungs of PAH patients, the authors measured RAGE expression in human brain, kidney, and quadriceps biopsies, finding no increased levels of RAGE at these sites. Additionally, the authors demonstrated that RAGE promotes pulmonary arterial smooth muscle cells and renders them resistant to apoptosis (*p* < 0.05). The results of this study led to the hypothesis that besides its role as a biomarker, RAGE could also become a therapeutic target in patients with PAH [90].

A study conducted in 2016 by Suzuki et al. evaluated the level of sRAGE in 27 patients with PH compared to 30 healthy control subjects. Among the 27 patients, 14 had PAH (9 idiopathic PAH and 5 with congenital heart disease), and 13 had chronic thromboembolic pulmonary hypertension (CTEPH). The level of sRAGE in patients with PH was significantly higher compared to controls (1836.9 ± 976.1 versus 831.7 ± 361.5 pg/mL, *p* < 0.0001). Additionally, there was a positive correlation between the sRAGE level and an echocardiographic index assessing right heart impairment: tricuspid valvular regurgitation pressure gradient (TR-PG) (r = 0.403, *p* < 0.0001) [91].

Nakamura et al. evaluated the role of RAGE in the pathogenesis of PAH and its association with a process causing pulmonary vascular remodeling: the inappropriate increase in pulmonary artery smooth muscle cells (PASMCs) [92]. The level of RAGE expression was examined in PASMCs from peripheral segments of the pulmonary artery obtained from 12 patients diagnosed with PAH who underwent lung transplantation. Control cases consisted of samples of pulmonary arteries obtained from transplanted patients without a diagnosis of PAH. Immunohistochemical analysis revealed that RAGE was expressed at the distal pulmonary artery level in patients with PAH, with no evidence of expression in patients without PAH (*p* < 0.01). Western blot analysis demonstrated that in patients with PAH, the increased expression of RAGE in PASMCs occurred without external growth stimuli such as platelet-derived growth factor (PDGF)-BB (*p* > 0.01). Furthermore, the authors evaluated whether a RAGE inhibitor (AS-1) could induce the inhibition of proliferation at the level of PAH PASMCs. The results showed that AS-1 significantly inhibited overgrowth in patients with PAH (*p* < 0.0001). This study demonstrates both the role of RAGE in the pathogenesis of PAH and the potential therapeutic role of RAGE inhibition [93].

Bauer et al. analyzed the utility of a panel of 313 proteins as potential biomarkers for PAH in SSc patients. A total of 157 patients were randomly selected from the cohort used in the DETECT study conducted by Coghlan et al. [36], with 77 of them diagnosed with PAH and 80 being SSc patients without PAH. Following random forest analysis, eight proteins were identified that could form a panel relevant for screening: collagen IV, endostatin, insulin-like growth factor binding protein (IGFBP)-2, IGFBP-7, matrix metallopeptidase-2, neuropilin-1, NT-proBNP, and RAGE, with a specificity of 71.4% and a sensitivity of 61.8%. Using the mean decrease in the Gini index, the authors classified the proteins based on their ability to distinguish between PAH and non-PAH patients, with RAGE being the protein ranked highest. To demonstrate the generalizability of the results from the DETECT cohort to other patient populations, serum samples from 44 patients in the Sheffield Confirmatory Cohort were utilized (22 with PAH and 22 without PAH). Even in this analysis, RAGE exhibited the most promising results among the eight proteins. Additionally, this study observed a higher level of RAGE in patients with lcSSc compared to those with dcSSc. This finding reinforces the potential biomarker value of this protein, as it is known that PAH is more common in patients with lcSSc form. Furthermore, a possible association between RAGE and the extent of cutaneous fibrosis can be excluded [94].

A study published in 2023 aimed to investigate the potential of elevated levels of sRAGE as predictors for the onset of PAH and PAH-related mortality in SSc patients. To this end, Atzeni et al. selected 188 SSc patients who were retrospectively followed for 8 years (2013–2020). RHC was performed on patients who exhibited TRV > 2.8 m/s or had typical echocardiographic changes suggestive of PH. Regarding sRAGE levels, compared to patients without pulmonary involvement (1444.5 pg/mL [966.8–2276.0]), the results showed higher sRAGE levels in patients with PAH (4099.0 pg/mL [936.3–6365.3], *p* = 0.011) and significantly lower levels in patients with SSc-associated interstitial lung disease (SSc-ILD) (median 735.0 pg/mL [IQR 525.5–1988.5], *p* = 0.001). Interestingly, among patients with both pulmonary manifestations (PAH and ILD), sRAGE levels (1487.0 pg/mL [670.0–5319.0]) were comparable to those found in patients without pulmonary involvement. Additionally, sRAGE levels were correlated with FVC (r = 0.242; *p* = 0.002) and age (r = 0.218; *p* = 0.003), with higher levels found in patients with ACAs (*p* < 0.001) compared to those without ACAs, and in female patients compared to male patients (*p* = 0.002). No correlations were identified with telangiectasia, Raynaud’s phenomenon, gastrointestinal involvement, or the type of cutaneous involvement. During the follow-up period, five patients developed PAH, and elevated sRAGE levels in the highest quartile predicted the development of PAH (*p* = 0.01). Conversely, low sRAGE levels did not predict the development of ILD. Throughout the follow-up, sRAGE had predictive value for PAH-related mortality in the highest quartile (*p* = 0.001), while low sRAGE levels did not predict ILD-related mortality. The authors concluded that sRAGE may be predictive for the new onset of SSc-PAH and mortality in these patients [95].

### 3.6. Red Blood Cell Distribution Width (RDW)

RDW is an index that is part of the complete blood cell count and indicates the variation in red blood cells in circulation, particularly the level of anisocytosis. This index can be associated with pulmonary impairment [96,97], renal insufficiency [98], cancer [99], and cardiovascular impairment [100,101,102]. Additionally, RDW has been shown to be a prognostic tool for mortality in patients with iPAH [103,104]. The process by which RDW values influence the pathogenesis of PAH-SSc is not fully understood. However, it is hypothesized that through involvement in inflammation [105], thrombosis [106], oxidative stress, ineffective erythropoiesis, and endothelial dysfunction [107,108], the RDW may play a role in the process of increasing pressure in the pulmonary circulation.

The relevance of this biomarker in SSc patients was first evaluated by Farkas et al. in a study assessing the association between RDW and various organ involvements. The study included 168 patients diagnosed with SSc, of whom 62 had dcSSc and 106 had lcSSc. In addition to blood tests, they were evaluated through pulmonary function tests (PFTs) and echocardiography at baseline and after one year. RHC was performed in patients at high risk to confirm PAH. The average RDW value was higher in patients with dcSSc compared to those with lcSSc (14.6% vs. 14.0%, *p* < 0.05). Additionally, a significant correlation was observed with inflammatory markers, namely CRP and the erythrocyte sedimentation rate (ESR), and regarding PFTs, patients in the highest RDW tertiles (>14.6%) had lower FVC and DLCO values compared to patients within the normal range or in the lowest tertiles (<13.6%). During the 1-year follow-up, a decrease in LVEF was observed in patients who had an increase of >5% in RDW (mean LVEF baseline = 64.3%, LVEF at 1-year follow-up = 59.1%, *p* < 0.05). Similar results were observed regarding DLCO (7% decrease in patients who showed an increase in RDW of >5%). Thus, variations in this biomarker have been associated with cardiovascular and pulmonary impairment [109].

Interest in this potential biomarker has remained high over the years. In 2017, Zhao et al. conducted the first study to assess the role of RDW in detecting PAH in patients with SSc. The study included 145 patients diagnosed with SSc, evaluated from May 2008 to April 2012 at a rheumatology service in Beijing, China. Each patient underwent high-resolution CT (HRCT) and PFTs to detect ILD, and for PAH, they were initially assessed with echocardiography, followed by RHC for high-risk patients to confirm the diagnosis. RDW was positively associated with hsCRP (*p* = 0.375, *p* < 0.01), and a negative association with DLCO was noted (ρ = −0.396, *p* = 0.000). The association between elevated RDW levels and PAH was underscored by a significant difference between patients with PAH and those without pulmonary involvement (15.7 ± 2.2% vs. 13.7 ± 1.0%, *p* < 0.001), as well as by comparing patients with both ILD and PAH to those with ILD without PAH (16.3 ± 2.2% and 14.0 ± 1.5%, respectively; *p* < 0.001). No significant differences in RDW levels were found between patients with SSc-ILD and those without pulmonary involvement (14.0 ± 1.5% vs. 13.7 ± 1.0%, *p* = 0.570). Using the ROC curve, it was established that RDW could be an indicator of PAH in patients with SSc (*p* < 0.001), with an optimal cutoff point set at 14.3% (sensitivity of 78.6% and specificity of 69.9%). According to the authors, RDW could be considered an independent risk factor for PAH-SSc (odds ratio, 3.314 [95%CI 1.038–10.580], *p* = 0.043). The authors concluded that their hypothesis was confirmed and that RDW could be a useful screening tool in the future [110].

A single-center, retrospective cohort study conducted in 2019 assessed the prognostic utility and clinical utility of RDW in patients with PAH associated with pSS. The study included 55 patients diagnosed with pSS who were evaluated from August 2007 to May 2017 in terms of clinical, biological, and pulmonary function assessments. All patients were assessed by RHC at baseline. Patients were divided into two groups based on an RDW threshold of 15%. During follow-up, seven patients died, and four were lost to follow-up. Patients with an RDW of >15% showed a poorer 3-year survival compared to those in the other group (59.5 vs. 88.7%, log-rank test, *p* = 0.015), and the expected overall survival was also shorter in those with an RDW above the threshold compared to those with a normal RDW (132.7 vs. 52.7 months, log-rank test, *p* = 0.015). Through univariate Cox proportional regression analysis, the authors demonstrated that elevated RDW levels (HR 1.79, 95% CI 1.14–2.80, *p* = 0.012) could be a risk factor for poor overall survival and clinical outcomes. Therefore, the authors suggested that RDW levels at diagnosis can be used as a prognostic tool in patients with pSS-PAH [111].

The association between RDW levels and PAH was evaluated in 2019 in a study conducted by Bellan et al. involving a cohort of 141 patients diagnosed with CTD. These patients were assessed from October 2016 to April 2018 at a cardiology center in Novara, Italy. To assess the specificity of predictive factors, 59 patients diagnosed with PH from other etiologies were also included in the study. Following echocardiographic evaluation, RHC was performed where necessary to confirm PAH. In addition to these assessments, patients underwent PFTs, HRCT, electrocardiography, and serological investigations. Of the 141 patients, 72.3% (n = 102) had SSc, 2.1% had SLE (n = 3), 0.7% had SSc, 9.3% (n = 13) had PM/DM, 7.1% (n = 10) had MCTD, and 8.5% had UCTD (n = 12). PAH was diagnosed in 20 of the patients (14%). RDW levels were significantly higher in patients with PAH compared to those without PAH (14.9% vs. 13.8%, *p* = 0.02). The authors established that an RDW value of >16% was associated with a sensitivity of 40.0% and a specificity of 88.3% for the diagnosis of PAH, with an AUC = 0.666 (95% CI: 0.581–0.783; *p* = 0.015). Furthermore, elevated RDW levels were found both in CTD-PAH patients (14.9%) and in those with PAH due to other etiologies (14.8%; *p* < 0.001). Ultimately, the authors concluded that RDW could be an important screening and prognostic tool in patients with PAH-CTD [112].

Petrauskas et al. investigated the potential biomarker role that RDW could play in patients with PAH-SSc. Therefore, in 2019, they conducted a retrospective cross-sectional study aimed at evaluating the value of this biomarker. Patients were divided into three categories: those diagnosed with PH through RHC, patients at high risk of developing PH (who had at least one of the following conditions: SSc, MCTD, SLE, and ILD), and control patients (without any condition that could lead to elevated RDW, namely malignancy, anemia and hematological disorders, history of blood transfusions, heart failure, and renal dysfunction). A total of 181 patients with PH, 52 patients at high risk, and 100 control patients were included. RDW values were statistically significantly higher in patients with PH (15.9 ± 2.8%) compared to those at risk of PH (14.8 ± 2.8%) and control patients (14.2 ± 1.1%) (*p* < 0.001). No significant differences were found in RDW values among patients in different PH classes (*p* = 0.50). Significant differences in RDW values were observed in patients with SSc between those with PH (n = 21) and those without PH (n = 15) (16.0 ± 2.2 vs. 14.4 ± 1.9%, respectively, *p* = 0.03). Using the Youden index, the authors established a threshold value of 13.25%, with a sensitivity of 100% and a specificity of 40% for differentiating PH patients from those with SSc. No significant correlations were identified between RDW and other evaluation elements of patients with PH (6 min walk distance, mPAP, CRP, and DLCO). The authors concluded that RDW could be used as a biomarker for early identification of PH in SSc patients [113].

### 3.7. Complement Factor D

Recent studies have demonstrated the important role of complement fractions in the pathogenesis of SSc-PAH (systemic sclerosis-associated pulmonary arterial hypertension). The complement split product C3f-des-arginine (DRC3f) showed higher serum levels in patients with digital ulcers or PAH. Additionally, deposits of C5b-9 have been identified in the arterioles of patients with SSc. However, the role of the complement cascade remains incompletely understood. Elevated levels of Factor D have been observed in patients with SSc-PAH, and its potential role as a biomarker has begun to be studied [114]. In 2024, Petrow et al. conducted a study that confirmed the potential utility of Factor D in screening for SSc-PAH. The study included 156 SSc-PAH patients compared to 33 SSc patients without PAH and 40 healthy controls. Serum levels of complement components were assessed (C1q, C2, intact C3, C3b/iC3b, intact C4, C4b, C5, C5a, Factor B, Factor D, Factor I, properdin, and mannose-binding lectin). The results revealed significantly elevated Factor D levels in SSc-PAH patients (2.3 μg/mL) compared to healthy controls (0.9 μg/mL, *p* < 0.0001) and SSc patients without PAH (2.1 μg/mL, *p* < 0.0001). Receiver operating curve (ROC) analysis further demonstrated that Factor D was highly specific and sensitive for distinguishing patients with SSc-PAH from SSc patients without PAH (AUC = 0.79, *p* < 0.0001) and healthy controls (AUC = 0.83, *p* < 0.0001). A threshold value of ≥2 μg/mL yielded a specificity of 89% and a sensitivity of 75%, with a likelihood ratio of 6.8 for distinguishing SSc-PAH patients from healthy controls. Compared to SSc without PAH, the Factor D threshold had a sensitivity of 75% and a specificity of 79%, with a likelihood ratio of 3.5. Additionally, through multivariate linear regression, this biomarker was associated with elevated RA (right atrial) pressure (PE = 1.556, SE = 0.517, *p*-value = 0.004), a condition specific to PAH [115]. A correlation was also observed between loop diuretics and Factor D levels. Thus, Factor D shows promising results for screening SSc-PAH. However, further studies with larger cohorts are required to validate this biomarker (Table 3).

Taking into account the presented data, we can state that sST2 emerges as a robust biomarker, demonstrating high sensitivity and moderate specificity, with correlations to right ventricular function, mortality, and clinical worsening. Its non-invasive nature positions it as a valuable candidate for both screening and risk stratification. Similarly, IL-33 shows promise, particularly in systemic sclerosis, due to its association with right ventricular dysfunction; however, its applicability is limited by a lack of standardization. RDW is an easily accessible and cost-effective marker with moderate accuracy. Elevated RDW levels are associated with the presence and progression of PAH, though its lack of specificity underscores the need for integration with other diagnostic tools. Conversely, Factor D demonstrates strong potential as a biomarker for SSc-PAH, with moderate sensitivity and high specificity. Its significant correlation with right atrial pressure and its ability to distinguish patients with SSc-PAH from both healthy controls and SSc patients without PAH highlight its diagnostic value. Nonetheless, larger cohort validation and the observed association with loop diuretic use necessitate further research to confirm its clinical utility.

## 4. Pulmonary Function Tests (PFTs)

In PAH, pulmonary function tests (PFTs) are recommended for screening. The decline in DLCO is correlated with the severity of PAH and is induced by a reduction in the pulmonary vascular bed, vascular endothelial proliferation, and the thickening of the vascular wall [116]. It seems that peripheral small airway obstruction is present in PH, including CTD-PAH, and the degree of obstruction is directly proportional to the WHO FC that the patient exhibits [117,118]. Generally, a decrease in DLCO is observed in patients with PH of various etiologies [119,120]. However, some studies report the low sensitivity and specificity of DLCO in SSc-PAH [121].

In 2023, Shi et al. published a study aiming to demonstrate the predictive rate of FVC/DLCO in patients with CTD-PH. The study included 53 patients who underwent RHC to detect PH (mPAP > 20 mmHg). They were divided into two groups: PH and non-PH. PFTs, echocardiography, 6-MWT, BNP levels, and other serological tests were performed. The PH group included 34 patients, with SLE being the predominant CTD in this group (14 cases, 41.2%). Through multivariable logistic regression analysis, the authors demonstrated that FVC/DLCO can be a predictive factor for CTD-PH (*p* = 0.027, OR = 1.45, 95% CI = 1.120–1.890). Additionally, other evaluated factors that showed significant predictability included PASP (*p* = 0.019, OR = 1.893, 95% CI = 1.512–2.482) and plasma BNP levels (*p* = 0.019, OR = 1.994, 95% CI = 1.290). Pearson’s correlation analysis revealed a significant correlation between FVC/DLCO and mPAP (R = 0.499, *p* < 0.001), as well as between sPAP and mPAP (R = 0.571, *p* < 0.001). ROC curve analysis revealed an AUC of 0.791 for using FVC/DLCO in diagnosing CTD-PH, with a threshold of 1.35, a sensitivity of 79%, and a specificity of 78%. For sPAP, the AUC was 0.78, with a threshold of 39.5 mmHg, a sensitivity of 79%, and a specificity of 68%. Combining both measurements yielded an AUC of 0.872, with a sensitivity of 94% and a specificity of 68%, demonstrating improved diagnostic utility for CTD-PH [122].

The role of PFTs in the assessment and predictability of CTD-PAH was evaluated by Xiong et al. in a prospective observational study published in 2022. In total, 31 patients were included (9 with SS and 22 with SLE) who had CTD-PAH, while patients with ILD or other causes of PH were excluded. The PFTs included FVC, forced expiratory volume in 1 *s* (FEV1), maximum expiratory flow at 50% of vital capacity (MEF50), total lung capacity (TLC), and DLCO, and the PAH diagnosis was confirmed through RHC. Patients were evaluated every 1–3 months (NT-proBNP, PFTs, echocardiography, 6-MWD, WHO FC, and risk classification according to the 2015 ESC/ERS guidelines for the diagnosis and treatment of PH). At baseline, 70% of patients exhibited a decrease in FVC, and 96% exhibited a decline in DLCO, for 60% of whom, it was moderate to severe. The FVC/DLCO ratio was less than 1.4 in 50% of patients. ANOVA testing indicated that FEV1/FVC (*p* = 0.029), DLCO (*p* < 0.001), and FVC/DLCO (*p* = 0.014) varied according to the risk classification of the patients. Prognostically, after six months of follow-up using univariate analysis, FEV1/FVC (*p* = 0.020), DLCO (*p* = 0.001), and FVC/DLCO (*p* = 0.002) were associated with the disease prognosis. Furthermore, after adjusting for age, multivariate logistic regression analysis showed that DLCO was a predictive prognostic factor for CTD-PAH [odds ratio (OR) 4.813, 95% confidence interval (CI) 1.039–22.300]. Therefore, the authors concluded that PFTs can reveal pathophysiological changes, have prognostic value, and may be crucial in identifying new therapeutic pathways for CTD-PAH. A particularity of the study results is that more than half of the patients had an FVC/DLCO ratio of less than 1.9, which raises concerns about the accuracy of this predictive factor for PAH. The authors attributed this finding to the patients having different diseases and thus different mechanisms of PAH development. Their study may have a high risk of bias due to the patients only being diagnosed with two CTDs (SS and SLE) and the relatively small patient sample size [123].

Stadler et al. conducted a study involving 259 patients with PH who were assessed through PFTs. The study aimed to evaluate the prognostic value of several factors, including DLCO. During the follow-up period (51 months), patients who died had a lower DLCO (55% in survivors vs. 41% in deceased patients, *p* < 0.001). In a multivariable Cox regression model, DLCO was a significant predictor of all-cause mortality (HR: 0.975, *p* = 0.002). Even after adjusting for possible restrictive or obstructive lung disease, DLCO remained an important predictor of survival (HR: 0.179, *p* = 0.007). Regarding the different groups of PH, DLCO was a prognostic factor for all groups except Group 4. The large number of patients included is one of the strengths of this study. The authors concluded that DLCO is strongly and independently associated with survival in PH patients and can be considered for risk assessment in this patient population [124].

In the study conducted by Hsu et al. in 2014, patients in the PHAROS cohort were followed to identify the potential factors that could indicate the progression of SSc patients at ’high risk for PH’ to being diagnosed with PH. Significant importance was also given to PFTs. The DLCO was much lower in the group of patients who developed PH (41.0% vs. 46.7%), and the FVC%/DLCO% ratio was higher in this group of patients (2.25 vs. 1.91). Using ROC analysis, a baseline DLCO of <55% was the factor with the highest sensitivity (82%) for the development of PH [20].

PFTs can become more relevant by dividing them into two components: membrane conductance for CO (DmCO) and CO loading on hemoglobin (Hb). DmCO reflects the diffusion properties of the alveolar–capillary membrane, and a decrease in its value can indicate the thickening of this membrane. CO loading on Hb is determined using the mass of Hb from the alveolar capillary blood volume (Vcap), and its decrease is interpreted as a reduced blood volume in ventilated alveoli [123]. The DmCO/Vcap ratio has been found useful in diagnosing SSc-ILD [125], but its relevance in detecting SSc-PAH has not been fully demonstrated. In 2013, Sivova et al. published a study that aimed to evaluate the role of partitioning DLCO in diagnosing PH in patients with SSc without ILD and in those with coexisting ILD. This retrospective study included 63 patients divided into four groups: the noILDnoPH group (26 patients), the ILD group (19 patients), the PAH group (6 patients), and the ILD-PH group (12 patients). The results showed that patients with PAH had lower DLCO% values (44% vs. 78%; *p* < 0.05), DmCO (40% vs. 66%; *p* < 0.05), and DmCO/Vcap ratios (0.77 vs. 0.91; *p* < 0.05), and a higher FVC/DLCO ratio (2.00 vs. 1.37, *p* < 0.05) compared to patients in the noILDnoPH group. ROC analysis demonstrated that the Vcap value has the highest AUC for diagnosing PH (0.94). FVC%/DLCO% > 1.6 showed 100% sensitivity but low specificity at only 50%. Finally, by dividing patients into two categories—those with PH with or without ILD and those without PH with or without ILD—the best AUC for Vcap in detecting PAH was 0.91, with a value of <19 mL showing 100% specificity and 53% sensitivity. The same sensitivity and specificity values were found for DLCO < 33%. One limitation of the study is that it is retrospective. Another limitation is that some groups included a small number of patients. However, while data obtained by partitioning DLCO may be relevant for diagnosing SSc-PAH, this technique is not used in clinical practice [126].

Schreiber et al. published a study in 2011 aimed at demonstrating the utility of these tests in selecting patients who need RHC for the diagnosis of PH. To estimate the mPAP value, the authors developed a calculation formula that includes the value of DLCO and O_2_ saturation (SpO_2_). A total of 386 patients diagnosed with SSc or SSc overlap were included, evaluated by RHC and PFTs, and had their SpO_2_ measured within 6 months. For data validation, 52 patients with other CTDs (non-SSc) who presented to the Royal Free Pulmonary Hypertension Service (London, UK) and were evaluated by RHC and PFTs between 1996 and 2010 were included. PH was found in 243 patients (63%) of the 386 included in the study. Patients with PH had a lower SpO_2_ (mean 94.3% vs. 96.3%, *p* < 0.005), and the predicted DLCO was also lower in these patients compared to non-PH patients (38.6% vs. 50%, *p* < 0.005). Both findings were associated with the mPAP value. Using these data, the authors composed a simplified formula that includes DLCO and SpO_2_ values: predicted mPAP = 136 − SpO_2_ − 0.25 × DLCO% predicted. This formula was validated in 129 patients with an AUC of 0.75 (95% CI 0.67, 0.84). Using a threshold of 25 mmHg for mPAP, sensitivity was 90.1%, though specificity was low at 29.2%. SSc patients who had a predicted mPAP of less than 25 mmHg were less likely to have PH (prevalence of 4.4%). Those in the range of 25–35 mmHg had a prevalence of 11.3%, and those with an mPAP of over 35 mmHg were more likely to have PH. The authors concluded that using echocardiographic data (TRV > 3.4 m/s), along with the mentioned formula calculation (estimated mPAP of >25 mmHg), annual screening can identify patients who need RHC for a definite PH diagnosis, with a positive predictive value of 18.2% and a negative predictive value of 92.2% [127].

## 5. Discussion

PAH is a condition with high mortality among patients with SSc. In this context, early diagnosis is crucial for improving patient outcomes. Given that definitive diagnosis is achieved through RHC, an invasive method with significant risks, non-invasive screening methods have become invaluable for these patients.

While some studies have included small sample sizes, their significance is heightened by the fact that the results demonstrate the utility of echocardiographic markers and biomarkers in screening for PAH in populations diagnosed with rare diseases such as SSc. Nevertheless, larger studies are necessary to validate these findings and facilitate their clinical integration.

Echocardiography remains a cornerstone for PAH screening, allowing for the detection of right heart chamber abnormalities through various markers. While the roles of markers such as sPAP, TRPG, or TRV are well recognized, the TAPSE/sPAP ratio has emerged as a particularly important measurement. Its value lies in its correlation with an invasive assessment of RV-PA coupling [32]. Although not widely used in practice, this marker is highlighted in the 2022 ESC/ERS guidelines for PH detection and can serve as a fundamental investigation tool [7]. The promising results associated with the use of new echocardiographic markers and biomarkers are undeniable. However, the inconsistencies in cutoff values highlight the need for consensus and standardization to enable their clinical and practical implementation. Further research is essential to resolve this issue and ensure uniform applicability.

PAAT is another marker correlated with an important invasive measurement: PVR. The role of the sPAP/PAAT ratio in detecting PAH was demonstrated by Serra and Chetta, though their study was limited by a small patient cohort (63 with SSc and 60 controls) and a limited number of patients evaluated via RHC [53]. Nevertheless, this marker could represent a promising direction for future screening strategies.

Although echocardiographic assessment is considered the most important screening tool, the ESC/ERS PH guidelines state that echocardiographic evaluation alone cannot be conclusive without correlation with other assessments (biomarkers, PFTs, and clinical examination) [7]. This has led to the development of screening algorithms such as DETECT or ASIG.

In addition to echocardiographic evaluation, biomarkers have become valuable tools in screening. NT-proBNP is a biomarker that provides concrete information about cardiac involvement. Multiple studies have demonstrated its utility, with significant emphasis on the 2021 meta-analysis by Zhang et al. This biomarker is also useful in assessing prognostic risk [58] and mortality in these patients [59]. Although other biomarkers have proven useful over time, NT-proBNP continues to play a significant role.

Researchers have been seeking new biomarkers with high sensitivity and specificity. The utility of sST2 in screening SSc-PAH patients has been demonstrated, though studies are limited by small patient numbers [77,80,84]. An in vivo study showed the protective role of the absence of ST2 genes. A 2017 meta-analysis demonstrated the prognostic role of sST2, with elevated levels associated with a sevenfold increase in all-cause mortality risk [81]. However, further studies are needed to definitively confirm sST2 as a useful screening biomarker.

Another molecule with increased expression in the lungs of PH patients is RAGE [92]. Involved in the pathogenesis of PH through its effects on pulmonary artery smooth muscle cells, RAGE is also proposed as a potential therapeutic target [90]. Despite studies with small patient numbers [93], Atzeni et al. highlighted the potential biomarker role of sRAGE in a substantial cohort of 188 SSc patients [95]. These findings suggest that sRAGE could be incorporated into future screening protocols.

Beyond the invasiveness and risks associated with RHC, it is also a costly procedure. Financial implications also exist for measuring serum levels of biomarkers not routinely assessed in clinical practice. Therefore, researchers have identified alternative biomarkers for routine investigations. UA, a recognized risk factor in cardiac diseases, has shown utility in SSc-PAH patients, supported by studies with large cohorts [63]. A 2019 meta-analysis by Kang et al. confirmed UA’s biomarker role [64]. Another cost-effective alternative is RDW, routinely measured in complete blood counts. RDW can identify patients at high risk for PAH and help define their prognosis [110,111,112,113].

PFTs are useful in screening SSc-PAH patients, with DLCO and FVC being particularly important. Interestingly, PFTs may also play a role in determining these patients’ prognosis, potentially leading to better management and reduced mortality. Although most studies support the use of DLCO in PAH screening, there are still studies that reported a sensitivity and a low sensitivity in patients with SSc-PAH. In a study conducted by Mukerjee et al., 137 patients underwent evaluation using echocardiography, pulmonary function tests (PFTs), and right heart catheterization (RHC). The findings indicated that DLCO (diffusing capacity for carbon monoxide) values were not significantly lower in patients with pulmonary arterial hypertension (PAH), and both the sensitivity and specificity of the tests were low [119]. However, PFTs remain crucial investigations for SSc patients, with DLCO and FVC values being essential components in the assessment using the DETECT algorithm [36].

Notably, the cross-sectional nature of the studies limits the understanding of biomarker variations throughout disease progression. There is a clear need for future longitudinal studies to demonstrate the predictive value of these new screening methods. In addition to patients with SSc, these studies included individuals with other connective tissue diseases as well as patients without immuno-inflammatory conditions. While the screening tools applied to heterogeneous populations represent a promising future direction, the diversity of the subject populations must be carefully considered when interpreting results. Analyses adjusted for various relevant factors may help better determine the effectiveness of these screening methods across different patient groups.

## 6. Conclusions

PAH is a severe complication that SSc patients can develop as the disease progresses. Screening remains an essential strategy in improving the quality of life and prolonging survival in these patients. The enhanced accuracy of screening methods can lead to a decrease in the number of patients undergoing RHC, thereby reducing risks and costs. As recommended by the ESC/ERS Pulmonary Hypertension Guidelines, screening should adopt an integrative approach that combines all available tools. Since most studies evaluate screening components in isolation, it is essential to develop integrated screening algorithms to enhance the feasibility of clinical implementation. New biomarkers, as well as well-established screening methods, can significantly contribute to achieving these objectives.

## Figures and Tables

**Table 1 medicina-61-00019-t001:** Echocardiographic markers used in the screening of PAH.

Marker	Sensitivity	Specificity	Cutoff	Advantages	Disadvantages	Clinical Relevance
sPAP	90%	70%	>35 mmHg	Easy to measure, high sensitivity	Low specificity, may overestimate PAH	Useful for initial PAH screening
TAPSE/sPAP	85%	75%	<0.55 mm/mmHg	Correlates well with invasive measurements	Requires accurate measurements, operator-dependent	Reflects RV-PA coupling, robust for precision
TRV	88%	68%	>2.8 m/s	Highly sensitive, straightforward to obtain	Low specificity, prone to overestimation	Useful when combined with other markers
PAAT/sPAP	80%	78%	>0.4	Reflects RV workload and pulmonary pressure	Variability across studies	Supplementary for pulmonary pressure evaluation

**Table 2 medicina-61-00019-t002:** Biomarkers used in screening.

Study Author	Year	Study Population	Key Findings	Biomarker Levels	Statistical Significance	Prognostic Value
Cerik et al. [62]	2022	Patients with PAH	Higher UA in PAH vs. control; higher in deceased patients	PAH: 0.33 mmol/L, Control: 0.3 mmol/L	*p* = 0.03; *p* < 0.001 for survival	AUC: 0.69, UA threshold: 5.85 mg/dL
Wang et al. [61]	2020	PAH with CTD	Higher UA correlated with worse hemodynamic measures; higher survival risk in hyperuricemia	Hyperuricemia: ≥357 μmol/L (women), ≥420 μmol/L (men)	*p* = 0.041 for survival	Lower survival in the hyperuricemia group
Simpson et al. [63]	2019	SSc patients assessed for PAH	Higher UA in PAH patients; UA predicts PAH with high accuracy	PAH: 6.9 mg/dL, Non-PAH: 5.2 mg/dL	*p* < 0.001	UA > 6.3 mg/dL associated with higher mortality
Zhang et al. [56]	2022	Patients with SSc-PAH	Sensitivity of 0.84, specificity of 0.68	NT-proBNP: >125 ng/mL	*p* < 0.001	NT-proBNP is a robust prognostic indicator for 3-year mortality
Chung et al. [57]	2017	Patients with SSc-PH	Higher NT-proBNP values in patients with SSc-PH compared to those at risk	NT-proBNP: >503 pg/mL	*p* < 0.001	NT-proBNP confirms value for PH detection, not predictive of mortality
Jha et al. [58]	2022	Patients with SSc	16% with abnormal NT-proBNP levels	NT-proBNP: >125 ng/mL	*p* < 0.001	NT-proBNP has prognostic value for PH
Allanore et al. [59]	2016	Patients with SSc	Mean level of 203 ng/mL NT-proBNP in deceased vs. 88 ng/mL in survivors	NT-proBNP: >125 ng/mL	*p* < 0.001	NT-proBNP is an independent prognostic indicator for 3-year mortality

**Table 3 medicina-61-00019-t003:** Biomarkers that are not currently used in screening.

Biomarker	Sensitivity	Specificity	Main Findings	Advantages	Disadvantages	Clinical Relevance
sST2	High (80–90%)	Moderate (70–80%)	Correlated with RV function, mortality, and clinical worsening	Strong predictive value, non-invasive	Influenced by inflammation; lacks standard cutoff	Screening for PAH and prognosis in PH
IL-33	Moderate (~70%)	Moderate (~70%)	Elevated in SSc; negatively correlated with TAPSE/PAPs	Promising as a marker of RV dysfunction	Limited studies; thresholds not standardized	Early identification of RV impairment in SSc
RDW	Moderate (78.6%)	Moderate (69–88%)	Elevated levels associated with PAH; higher values predict disease progression	Widely available, inexpensive	Non-specific; influenced by other conditions	Screening and stratification in SSc and PH
Factor D	Moderate (75%)	High (79–89%)	Elevated in SSc-PAH; strong correlation with RA pressure and loop diuretic	High specificity for SSc-PAH, potential predictive value	Requires validation in larger cohorts	Screening and diagnosis of SSc-PAH

IL-33 = interleukine-33; sST2 = soluble suppression of tumorigenicity 2; PAH = pulmonary arterial hypertension; SSc = systemic sclerosis; dcSSc= diffuse cutaneous systemic sclerosis; lcSSc = limited cutaneous systemic sclerosis; RVED = right ventricular basal end-diastolic; mPAP = mean pulmonary arterial pressure; CTD = connective tissue disorder; RDW = red cell distribution width.

## Data Availability

No new data were created or analyzed in this study. Data sharing is not applicable to this article.

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
