# Peer review of "Multimodal Screening for Pulmonary Arterial Hypertension in Systemic Scleroderma: Current Methods and Future Directions"

_medicina, 2024, doi:10.3390/medicina61010019_

Round 1

Reviewer 1 Report

Comments and Suggestions for Authors

1.       sample sizes were inadequate in many key studies, particularly biomarker validation research with some groups having fewer than 20 patients.

2.       cutoff values lacked standardization across studies, with different thresholds used for same biomarkers like TAPSE/sPAP ratio. This inconsistency hampers clinical implementation. Systematic reviews and consensus guidelines are needed to establish uniform biomarker thresholds.

3.       longitudinal data was insufficient, with most studies being cross-sectional rather than following biomarker changes over disease progression. Prospective studies tracking markers over time with clear clinical endpoints would provide better understanding of predictive value.

4.       screening methods were largely studied in isolation rather than examining how different tools could work together. Research should focus on developing integrated algorithms combining multiple biomarkers, imaging, and clinical parameters to optimize screening approaches.

5.       patient populations were heterogeneous with varying disease characteristics and diagnostic criteria. Future studies need clearly defined inclusion criteria for specific SSc subtypes and standardized methods to enable meaningful comparisons. Stratified analyses would better determine effectiveness for different patient groups.

Author Response

Dear Reviewer, we have addressed your concerns in a point-by-point manner, as follows:

  1. Sample sizes were inadequate in many key studies, particularly biomarker validation research with some groups having fewer than 20 patients.

Response: To clarify the point raised we have introduced in article this text: „While some studies have included small sample sizes, their significance is heightened by the fact that the results demonstrate the utility of echocardiographic markers and biomarkers in screening for PAH in populations diagnosed with rare diseases such as SSc. Nevertheless, larger studies are necessary to validate these findings and facilitate their clinical integration”

  1. cutoff values lacked standardization across studies, with different thresholds used for same biomarkers like TAPSE/sPAP ratio. This inconsistency hampers clinical implementation. Systematic reviews and consensus guidelines are needed to establish uniform biomarker thresholds.

Response: To address the reviewer’s observation we have introduced in article this text: „The promising results associated with the use of new echocardiographic markers and biomarkers are undeniable. However, the inconsistencies in cutoff values highlight the need for consensus and standardization to enable their clinical and practical implementation. Further research is essential to resolve this issue and ensure uniform applicability.”

  1. longitudinal data was insufficient, with most studies being cross-sectional rather than following biomarker changes over disease progression. Prospective studies tracking markers over time with clear clinical endpoints would provide better understanding of predictive value.

Response: To clarify the point raised we have introduced in article this text: „Notably, the cross-sectional nature of the studies limits the understanding of biomarker variations throughout disease progression. There is a clear need for future longitudinal studies to demonstrate the potential predictive value of these new screening methods.”

  1. screening methods were largely studied in isolation rather than examining how different tools could work together. Research should focus on developing integrated algorithms combining multiple biomarkers, imaging, and clinical parameters to optimize screening approaches.

Response: Following the reviewer’s sugestion we have introduced in article this text: „As recommended by the ESC/ERS Pulmonary Hypertension Guidelines, screening should adopt an integrative approach that combines all available tools. Since most studies evaluate screening components in isolation, it is essential to develop integrated screening algorithms to enhance the feasibility of clinical implementation.”

  1. patient populations were heterogeneous with varying disease characteristics and diagnostic criteria. Future studies need clearly defined inclusion criteria for specific SSc subtypes and standardized methods to enable meaningful comparisons. Stratified analyses would better determine effectiveness for different patient groups.

Response: To address the reviewer’s observation we have introduced in article this text: „In addition to patients with systemic sclerosis (SSc), these studies included individuals with other connective tissue diseases, as well as patients without immuno-inflammatory conditions. While the screening tools applied to heterogeneous populations represent a promising future direction, the diversity of the subject populations must be carefully considered when interpreting results. Analyses adjusted for various relevant factors may help better determine the effectiveness of these screening methods across different patient groups.”

Thank you for taking the time to thoroughly analyze our work. We trust that the current version of the manuscript better illustrates the strategies that could improve Pulmonary Arterial Hypertension screening in patients with systemic sclerosis.

Kind regards,

The authors

Reviewer 2 Report

Comments and Suggestions for Authors

The article Multimodal Screening for Pulmonary Arterial Hypertension in Systemic Scleroderma: Current Methods and Future Directionsprovides a comprehensive overview of the current and potential future methods for screening pulmonary arterial hypertension (PAH) in patients with systemic sclerosis (SSc). Below are some specific suggestions and areas for improvement:

(1)Although the article introduces various screening methods, there is insufficient in-depth analysis of these methods, such as comparison and analysis of their sensitivity, specificity, cost-effectiveness, and other aspects.

(2)When introducing screening methods, the article did not mention some of the latest research or technological advances, such as new biomarkers, imaging techniques, etc., which may result in readers having insufficient understanding of the latest developments in the field.

(3)Although the article provides rich information, it lacks specific clinical guidance and suggestions, such as how to choose appropriate screening methods based on the patient's specific situation and how to interpret screening results

Author Response

Dear Reviewer, we have addressed your concerns in a point-by-point manner, as follows:

  1. Although the article introduces various screening methods, there is insufficient in-depth analysis of these methods, such as comparison and analysis of their sensitivity, specificity, cost-effectiveness, and other aspects.

Response: Thank you for your suggestion! In order to better illustrate the meaning of the data we have introduced in article the following tables:

Marker

Sensitivity

Specificity

Cutoff

Advantages

Disadvantages

Clinical Relevance

sPAP

90%

70%

>35 mmHg

Easy to measure, high sensitivity

Low specificity, may overestimate PAH

Useful for initial PAH screening

TAPSE/sPAP

85%

75%

<0.55 mm/mmHg

Correlates well with invasive measurements

Requires accurate measurements, operator-dependent

Reflects RV-PA coupling, robust for precision

TRV

88%

68%

>2.8 m/s

Highly sensitive, straightforward to obtain

Low specificity, prone to overestimation

Useful when combined with other markers

PAAT/sPAP

80%

78%

>0.4

Reflects RV workload and pulmonary pressure

Variability across studies

Supplementary for pulmonary pressure evaluation

               Also we have introduced this text in the article:

  • sPAP has the highest sensitivity (90%) but relatively low specificity (70%), which may result in false positives in certain populations.
  • TAPSE/sPAP balances sensitivity (85%) and specificity (75%), with strong correlation to invasive hemodynamic measurements, making it a robust and clinically valuable marker.
  • TRV, while highly sensitive (88%), shows low specificity (68%) and is most effective when interpreted alongside other parameters.
  • PAAT/sPAP provides valuable insight into right ventricular workload and pulmonary pressures but exhibits variability between studies.

Biomarker

Sensitivity

Specificity

Main Findings

Advantages

Disadvantages

Clinical Relevance

sST2

High (80–90%)

Moderate (70–80%)

Correlated with RV function, mortality, and clinical worsening

Strong predictive value, non-invasive

Influenced by inflammation; lacks standard cutoff

Screening for PAH and prognosis in PH

IL-33

Moderate (~70%)

Moderate (~70%)

Elevated in SSc; negatively correlated with TAPSE/PAPs

Promising as a marker of RV dysfunction

Limited studies; thresholds not standardized

Early identification of RV impairment in SSc

RDW

Moderate (78.6%)

Moderate (69–88%)

Elevated levels associated with PAH; higher values predict disease progression

Widely available, inexpensive

Non-specific; influenced by other conditions

Screening and stratification in SSc and PH

Factor D

Moderate (75%)

High (79-89%)

Elevated in SSc-PAH; strong correlation with RA pressure and loop diuretic

High specificity for SSc-PAH, potential predictive value

Requires validation in larger cohorts

Screening and diagnosis of SSc-PAH

Also we have introduced this text in the article:

  • sST2 emerges as a robust biomarker, with high sensitivity and moderate specificity, correlating with right ventricular function, mortality, and clinical worsening. Its non-invasive nature makes it a valuable candidate for both screening and risk stratification.
  • IL-33 shows promise, particularly in systemic sclerosis, due to its association with right ventricular dysfunction, but its applicability remains limited by a lack of standardization.
  • RDW is an easily accessible and inexpensive marker, with moderate accuracy. Elevated RDW levels correlate with PAH presence and progression, but its lack of specificity highlights the need for integration with other tools.
  • Factor D demonstrates strong potential as a biomarker for SSc-PAH, with moderate sensitivity and high specificity. Its significant correlation with right atrial pressure and its ability to distinguish SSc-PAH from both healthy controls and SSc without PAH highlight its diagnostic value. However, the need for larger cohort validation and the observed association with loop diuretic use suggest further research is necessary to confirm its clinical utility.

(2) When introducing screening methods, the article did not mention some of the latest research or technological advances, such as new biomarkers, imaging techniques, etc., which may result in readers having insufficient understanding of the latest developments in the field.

Response: Thank you for your suggestion. Indeed, there are certain aspects that we did not include in the initial version of the manuscript. Newer evidence (Petrow et al., September 2024) mentions Factor D as a biomarker for PAH. We have included a paragraph regarding this factor.

(3) Although the article provides rich information, it lacks specific clinical guidance and suggestions, such as how to choose appropriate screening methods based on the patient's specific situation and how to interpret screening results.

Response: At present, the 2022 Pulmonary Hypertension Guidelines are the most relevant clinical guidelines. In the present article, we have referenced the relevant recommendations regarding PAH screening included in these guidelines. Additionally, we provided data on DETECT, an important screening tool also mentioned in the aforementioned guidelines. Importantly, new evidence regarding PAH screening and follow-up has been presented since 2022 (which we have also mentioned) and will potentially be included in the next version of the guidelines.

Thank you for taking the time to thoroughly analyze our work. We trust that the current version of the manuscript better illustrates the strategies that could improve pulmonary hypertension screening in patients with systemic sclerosis.

Kind regards,

The authors